# Unraveling the Complexity of Childhood Polycystic Kidney Disease: A Case Study of Three Sisters

**DOI:** 10.3390/children10101700

**Published:** 2023-10-17

**Authors:** Ivana Trutin, Lea Oletić, Tamara Nikuševa-Martić

**Affiliations:** 1Department of Pediatrics, Sestre Milosrdnice University Hospital Center, 10000 Zagreb, Croatia; ivana.trutin@gmail.com (I.T.); lea.oletic@kbcsm.hr (L.O.); 2Department of Biology, School of Medicine, University of Zagreb, 10000 Zagreb, Croatia

**Keywords:** autosomal dominant polycystic kidney disease, children, mutation

## Abstract

Autosomal dominant polycystic kidney disease (ADPKD) is the most common hereditary kidney disorder, estimated to affect 1 in 1000 people. It displays a high level of variability in terms of onset and severity among affected individuals within the same family. In this case study, three sisters (4, 8, and 10 years of age) were suspected of having ADPKD due to their positive family history. While the two younger sisters aged 8 and 4 showed no disease complications and had normal kidney function, the oldest sister was found to have no dipping status on ambulatory blood pressure measurement (ABPM). Two of the sisters were discovered to have a *PKD1* mutation, while the third sister aged 8 was heterozygous for *TTC21B* c.1593_1595del, p. (Leu532del), which is a variant of uncertain significance (VUS). Environmental factors and genetic modifying factors are believed to contribute to the phenotypic variability observed in ADPKD. Identifying and understanding potential genetic and environmental modifiers of ADPKD could pave the way to targeted treatments for childhood ADPKD.

## 1. Introduction

Autosomal dominant polycystic kidney disease (ADPKD) represents the most prevalent genetic kidney disorder, arising from *PDK1* and, less frequently, *PDK2* gene mutations [1]. The genes responsible for encoding polycystin 1 and 2 constitute the calcium channel complex within the primary cilia of renal tubular epithelium. It is theorized that mutations in either the *PDK1* or *PDK2* gene lead to cyst formation in different organs, arising from disruptions in the structure or function of primary cilia [2]. While several family members are often affected, symptoms typically manifest during adulthood. However, the disease is now more frequently identified in children, primarily due to a positive family history or through the evaluation of hypertension or proteinuria [3]. Discovering a mutation in a family does not enable forecasting the disease’s progression due to widely recognized but inadequately understood intrafamilial variability [4]. The case presented here involves three sisters with a clinical presentation of bilateral cystic kidney disease. The sisters aged 10 and 4 had been confirmed with the same pathogenic variant, *PKD1* c.11803del, p. (Ala3935Pros*10), which is absent in gnomAD, while the third sister aged 10, with the same clinical presentation, was heterozygous for *TTC21B* c.1593–1595del, p. (Leu532del), which is a variant of uncertain significance (VUS). ADPKD stands as the most prevalent heritable multifocal cystic disease encountered in clinical practice. Typically, it is diagnosed in individuals with a family history through the presence of significantly enlarged kidneys displaying multiple bilateral cysts, detectable via ultrasound, computed tomography scan, or magnetic resonance imaging. In most cases, the need for genetic testing is minimal. However, while ADPKD diagnosis is often straightforward, the possibility of misdiagnosis exists. Genetic testing is important for determining such mutations and, depending on this, additional diagnostics might be required. It is necessary to monitor renal function to prevent the development of other possible complications, especially in cases of such an early manifestation of the disease.

## 2. Case Description

The presented case is about three sisters (4, 8, and 10 years old) with normal perinatal anamnesis; the asymptomatic girls were diagnosed with polycystic kidney disease by a targeted examination due to a positive family history. Their father was monitored by his family doctor for bilateral cystic kidney disease, but refused more detailed hospital treatment and genetic analysis, even though his mother suffered from polycystic kidney disease discovered due to arterial hypertension. She underwent a kidney transplant at the age of 53 because of end-stage renal disease. Unfortunately, she was not genetically tested. Each of the abovementioned three sisters had an ultrasound scan revealing bilateral renal cysts, ranging in size from 3 × 2 mm to the largest, 20 × 10 mm, in a 10-year-old girl. The ultrasound finding of the urinary system described hyperechoic parenchyma of both kidneys, without corticomedullary differentiation. The 10-year-old and the 8-year-old sisters had normal ABPM (except no dipping status during the night was detected in the 10-year-old sister, without waking during the night), and the youngest one had normal blood pressure by home measurements. There was no indication of the need for therapy introduction. They were monitored for 10 months, at 3-month intervals (Table 1). The recommended intervals of proteinuria screening and ABPM pressure measurements are once a year, but we were surprised by the clinical picture in all three sisters, as well as the early presentation of the clinical picture of ADPKD, so we checked them more often at the beginning of the follow-up. Blood tests, screening for proteinuria, the glomerular filtration rate estimated by Schwartz (GFR), and serum creatinine were regularly monitored. Heart, abdomen, and urinary system ultrasounds were also done for each of them. Blood pressure, screening of proteinuria, and GFR were performed at each examination. There was no significant proteinuria, and no arterial hypertension, except for no dipping status on ABPM in the 10-year-old sister. The overall renal function was normal during monitoring in all three sisters; also, no hepatic, pancreatic, or splenic cysts were found with abdomen ultrasound or mitral valve prolapse with the heart ultrasound. Considering the mode of inheritance and the clinical presentation, ADPKD was assumed. Subsequently, a genetic analysis using the next-generation sequencing (NGS) method was performed for a total of 43 genes included in the cystic kidney disease panel. The analysis included sequence analysis and copy number variation analysis of the following genes: *ANKS6*, *CEP164*, *CEP290*, *CEP83*, *COL4A1*, *CRB2*, *DCDC2*, *DNAJB11*, *DZIP1L*, *EYA1*, *GANAB*, *GLIS2*, *HNF1B*, *IFT172*, *INVS*, *IQCB1*, *JAG1*, *LRP5*, *MAPKBP1*, *NEK8*, *NOTCH2**, *NPHP1*, *NPHP3*, *NPHP4*, *OFD1*, *PAX2*, *PKD1*, *PKD2*, *PKHD1*, *PRKCSH*, *RPGRIP1L*, *SDCCAG8*, *SEC61A1*, *SEC63*, *SIX5*, *TMEM67*, *TSC1*, *TSC2*, *TTC21B*, *UMOD*, *VHL*, *WDR19,* and *ZNF423*. *PKD1* is technically very challenging to analyze due to its large size, high GC-content, and duplication of the first 33 exons with a high degree of homology (90–99% identity) to 6 nearby pseudogenes (PKD1 P1–P6). A test includes copy number variant detection and analysis of clinically relevant deep intronic variants, and covers a wide range of differential diagnoses. The following performance metrics were demonstrated: overall high mean coverage (205×); 100% of the target nucleotides covered at least 20x with a mapping quality threshold of 20; 99.5% of target nucleotides covered at least 20x with mapping quality threshold of 40. Sequence analysis using the cystic kidney disease panel identified a heterozygous frameshift variant, *PKD1* c.11803del, p. (Ala3935Profs*10) in two sisters (aged 4 and 10). The variant was classified as likely pathogenic, based on the established association between the gene and the patients’ phenotype, the variant’s absence in control populations, and variant type (frameshift). The third sister aged 8 years was heterozygous for *TTC21B* c.1593_1595del, p. (Leu532del), which is VUS. The variant results in the deletion and protein coding length changes because of an in-frame variant in gene *TTC21B*. This variant is not located in a repeat region. Additionally, the variant is absent in gnomAD, a large reference population database which aims to exclude individuals with severe pediatric disease [5]. Due to suspicion of a negative result, we repeated the testing not just once but twice to exclude the possibility of error. The detection performance of this panel showed nuclear DNA sensitivity to SNVs 99.89%, indels 1–50 bps 99.2%, one-exon deletion 100% and five exons CNV 98.7%, and specificity >99.9% for most variant types. In two sisters aged 4 and 10, in whom the *PKD1* c.11803del, p. (Ala3935Pros*10) mutation was found, brain magnetic resonance imaging (MRI) was performed for intracranial aneurysms, which was not confirmed by examination. At this stage, there was no indication of any treatment, but further monitoring of patients is important to include antihypertensive and renoprotective therapy in time, with all symptomatic measures to prevent the development of end stage renal disease (ESRD).

## 3. Discussion

In recent years, our comprehension of the genetic underpinnings of polycystic kidney diseases has significantly advanced, primarily through the identification of two key genes associated with ADPKD, namely *PKD1* and *PKD2* [1,2]. However, it is important to note that the polycystic phenotype can result from various distinct genetic anomalies, and several potential *PKD* genes exist that need to be fully characterized. Moreover, the complexity of this condition is heightened by the fact that the genes responsible for expressing this phenotype may necessitate interactions with other genetic or environmental factors before the disease becomes clinically evident [7]. The *PKD1* gene encodes polycystin 1, a glycoprotein that along with polycystin 2 forms an integral membrane protein complex with a large N-terminal extracellular region, multiple transmembrane domains, and a cytoplasmic C-tail that functions as a regulator of calcium-permeable cation channels and intracellular calcium homeostasis [8,9]. The complex regulates multiple signaling pathways to maintain normal renal tubular structures such as cilium length and function. It is involved in fluid-flow mechanosensation by the primary cilium in the renal epithelium [9]. *PKD1* pathogenic mutations lead to ADPKD type 1, which is a highly penetrant multisystem condition characterized by the formation of bilateral renal cysts [10]. Renal manifestations of ADPKD commonly involve hypertension and renal insufficiency. It is noteworthy that about 50% of individuals with ADPKD develop ESRD by the age of 60 years and they make up 7–15% of patients on renal replacement therapy [11,12]. Apart from the kidneys, cysts can also develop in various other organs, such as liver, seminal vesicles, pancreas, and arachnoid membrane. Besides *PKD1* mutation, ADPKD is rarely caused by a *PKD2* mutation and in small number of cases by unknown or rare mutations in other locations, including *GANAB* and *DNAJB11* [13,14]. Interestingly, the apparent clinical prevalence of ADPKD has been estimated to be around 50% of the expected gene frequency [15]. This suggests that the initial genetic ‘trigger’ leading to the development of the polycystic phenotype is insufficient on its own to manifest as clinically recognizable disease. It is plausible that neighboring genes could influence the expression of the disease. Some of the key genetic modifying factors that have been studied in the context of ADPKD include polygenic interactions: ADPKD is considered a complex genetic disorder, and multiple genes may interact to influence the disease phenotype. Studies have suggested that modifier genes, not directly related to *PKD1* or *PKD2*, can have effects on ADPKD. These genes may affect cellular processes related to cyst growth and kidney function, impacting the disease’s course. An illustrative example of significant phenotype modulation through mutation in a second gene occurs when contiguous deletions affect both the *PKD1* and *TSC2* genes. Patients with these combined *PKD1*/*TSC2* deletions exhibit a notably more severe form of PKD compared to those with mutations in genes alone. Polycystin proteins are expressed on many tissues, including vascular smooth muscle cells and the endothelial cells that make up the vascular wall; therefore, the ADPKD is linked to intracranial aneurysms and aortic dilatation, and dissection with mitral valve prolapse occurs in up to 25% of affected individuals, along with abdominal wall hernias [16,17]. The prevalence of intracranial aneurysms is higher in those with a positive family history of aneurysms or subarachnoid hemorrhage than in those without such a family history [18]. There is a lot of discussion about whether to perform MRI in children suffering from ADPKD for the purpose of excluding intracranial aneurysms, and in case of negative family history for rupture of intracranial aneurysm. In rare cases with a positive family history and a strong desire to ease anxiety by screening, an individualized approach is justified [19]. Walker and Marlais suggest that an individual approach should be sought, and the families should be counselled about the potential of having an unruptured intracranial aneurysm [20]. Blood pressure monitoring, follow-up of renal function and ultrasound, as well as MRI angiography for screening of intracranial aneurysms in patients at high risk, is recommended [19]. In the case of our two sisters (ages 4 and 10), we found variant generates a frameshift in exon 43, resulting in a premature stop codon. *PKD1* c.11803del, p. (Ala3935Profs*10) was classified as likely pathogenic, based on the established association between the gene and the patients’ phenotype, the variant’s absence in control populations, and variant type (frameshift). Additionally, we had to consider the presence of a *TTC21B* variant, specifically c.1593_1595del, p. (Leu532del), in the eight-year-old third sister. This variant introduces the possibility that it could contribute to the observed phenotype. A single heterozygous mutation in the *TTC21B* gene has been linked to atypical nephronophthisis 12 (NPHP12), a rare form of hereditary nephropathy with a phenotype characterized by cystic renal disease. Unlike typical cases, atypical NPHP12 shows a milder onset and less severe clinical symptoms without developmental abnormalities [5]. The importance of genetic testing in childhood lies in differentiation of ADPKD from autosomal recessive polycystic kidney disease (ARPKD), contiguous *PKD1*-*TSC2* syndrome, or Meckel-Gruber syndrome. ADPKD may frequently show no symptoms in children, although it can manifest with hypertension or gross hematuria [3,21]. ABMP and heart ultrasound are critical in the assessment of ADPKD in children, as nearly one-third of children show exclusively nocturnal hypertension and as a consequence of possible hypertrophy of the left ventricle [6,22]. In adults, genetic testing for ADPKD is usually not performed because of the clearly established imaging diagnostic criteria and the technical challenges of sequencing *PKD1* [18]. Knowledge about genotype-phenotype correlation is increasing, so it is believed that genetic testing is important in adults with ADPKD, and especially in children with such an early manifestation of the disease, which was the case here. Long-term monitoring is important for the detection of hypertension development or chronic renal failure and timely therapeutic intervention. Protein excretion and hypertension should be monitored in children with ADPKD at the annual health care visit [19]. In the case of development of arterial hypertension, antihypertensive therapy is indicated according to the guidelines [6]. If proteinuria is present, angiotensin-converting enzyme inhibitors or angiotensin receptor blockers should be used as primary treatment [19]. A healthy lifestyle including physical activity, low dietary salt intake, and maintenance of normal weight are very important in all patients with autosomal dominant polycystic kidney disease [19]. Tolvaptan, an oral selective antagonist of the vasopressin V2 receptor, slows expansion of kidney volume and kidney function decline in adults with ADPKD [23]. Tolvaptan exhibited pharmacodynamic activity in pediatric ADPKD [24] but there are no clear instructions on the safety and indication of use in children suffering from ADPKD.

The main limitation of this paper is that there was no available genetic analysis of the father and father’s mother, which reduces the possibility of interpreting the findings. It is especially difficult to interpret the negative finding in the 8-year-old sister, who had a clinical presentation like that of her sisters with a pathogenic variant found. The other limitation is possible test restriction together with technical limitations, because a normal result does not rule out the diagnosis of a genetic disorder, since some DNA abnormalities may be undetectable by the applied technology. While genetic testing is a crucial diagnostic tool, the complexity of ADPKD extends beyond the identification of specific mutations. The clinical management of ADPKD patients, especially children, involves ongoing monitoring of renal function, blood pressure, and the development of complications. In this case, all three sisters exhibited normal renal function during their 10-month follow-up, with no significant proteinuria or hypertension detected. This was reassuring, but the importance of long-term surveillance cannot be overstated, as ADPKD is a progressive condition, and early intervention can be critical in preventing the development of ESRD.

In conclusion, our case raises suspicion that ADPKD can be misdiagnosed when other heritable multifocal cystic kidney diseases manifest without their characteristic features, and when there is a lack of genetic characterization among family members. This case study underscores the importance of genetic testing and long-term monitoring in children with ADPKD, particularly in families with a positive history of the disease. It highlights the challenges associated with intrafamilial variability and the interpretation of genetic findings. The identification of mutations allows for precise diagnosis and potential prognostic insights, although it is important to remember that the VUS requires further investigation to determine its clinical significance. As our understanding of ADPKD continues to evolve, further research into genetic modifiers and environmental factors may provide valuable insights into the management and treatment of this complex genetic kidney disorder.

## Figures and Tables

**Table 1 children-10-01700-t001:** Basic renal function results in sisters.

		Sister 1	Sister 2	Sister 3
Age (years)		10	8	4
BMI (kg/m^2^)		20.9/87.c.	18/77.c.	17/85.c.
Urea * (mmol/L)	At diagnosis	5.9	4	3.8
	3 months after	5.7	6.1	3.7
	9 months after	6	5.1	3.8
Serum creatinine ** (umol/L)	At diagnosis	35	30	28
	3 months after	35	32	30
	9 months after	37	31	35
24 h urine protein (mg/m^2^/h)	At diagnosis	5	4	4
	3 months after	4	4	4
	9 months after	5	4	4
ACR (mg/mL)	At diagnosis	3	<3	<3
	3 months after	3	<3	<3
	9 months after	3	<3	<3
24 h ABPM	At diagnosis	Non-dipping status BP < 90.c. [6]	Normal BP < 90.c. [6]	Home pressure measurements SBP/DBP < 90.c. [6]
	9 months after	Non-dipping status BP < 90.c. [6]	Normal BP < 90.c. [6]	Home pressure measurements SBP/DBP < 90.c. [6]
Urinary system ultrasound findings	At diagnosis	RK cyst 20 × 10 mm in the lower pole, several smaller cysts centrally; LK several cystic formations in the lower pole and centrally	RK cyst 8 × 12 mm in the lower pole, 2 smaller cyst centrally; LK several smaller cystic formations in the lower pole	RK cyst 6 × 6 mm in the lower pole, 2 smaller cyst cetrally; LK cyst 12 × 8 mm in the lower pole, several smaller cystic formations
	9 months after	No change compared to the findings at the time of diagnosis	No change compared to the findings at the time of diagnosis	No change compared to the findings at the time of diagnosis
Brain MRI angiography using the 3D TOF recording technique	At diagnosis	No aneurysmal expansions on the shown cerebral arteries, stenosis or AV malformations	N/A	No aneurysmal expansions on the shown cerebral arteries, stenosis or AV malformations
Treatment		None	None	None

BMI—body mass index; ACR—albumin/creatinin ratio; ABPM—ambulatory blood pressure monitoring; BP—blood pressure; SBP—systolic blood pressure; DBP—diastolic blood pressure; RK—right kidney; LK—left kidney; MRI—magnetic resonance imaging. * normal range 2.7–6.8; ** normal range 15–37.

## Data Availability

No new data were created or analyzed in this study. Data sharing is not applicable to this article.

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
