# Peer review of "Unraveling the Complexity of Childhood Polycystic Kidney Disease: A Case Study of Three Sisters"

_children, 2023, doi:10.3390/children10101700_

Round 1

Reviewer 1 Report

1) I would rather use term "asymptomatic" than "healthy" in this phrase:

"(...)healthy girls diagnosed with polycystic kidney disease by a targeted exami-48 nation due to a positive family history. "

2) Sentence: "Unfortunately, she was not genetically tested by a competent nephrologist." 

I would not pass judgement so easily, that this nephrologist was incompetent. Genetic testing is not necessary to diagnose ADPKD. I would state only, that sisters' grandmorther was not genetically testing. 

3) Why sister 3 had only home BP measurments and not ABPM? Was she too young for that? This study https://pubmed.ncbi.nlm.nih.gov/9438648/ showed that ABPM is a useful tool for the diagnosis and evaluation of hypertension in children under 6 years of age. Authors should explain this in more detail. 

4) First paragraph of the Discussion is basically the same what was already presented in Introduction. I would give up on this part.

5) Authors raised the significance of environmental factors for phenotype in ADPKD. What is the meaning of these factors in the presented case? Did Authors identify any differences in environmental factors among sisters? Diet? History of infections? Nephrotoxic drugs/other agents?

6) "Genetic testing played a pivotal role in elucidating the underlying genetic mutations responsible for ADPKD in this family." - this sentence is a complete truism, I would give this up. 

7) To make the case more educational I would add a paragraph about indications to start a treatment in children/adolescents with ADPKD.

This study could be cited in this paragraph - https://pubmed.ncbi.nlm.nih.gov/36719158/. 

Author Response

Dear Editor and Reviewers,

Thank you for your detailed and thoughtful review of our manuscript titled "Unraveling the Complexity of Childhood Polycystic Kidney Disease: A Case Study of Three Sisters." We have put in all our efforts to adress every question and comment provided by you. We have gone through and modified the discussion according to the instructions and we believe that thanks to your comments we have significantly improved the article. We are thankful for your constructive comments and we greatly appreciate your time and valuable input, which have undoubtedly strengthened our work.

Reviewer 1

1)I would rather use term "asymptomatic" than "healthy" in this phrase:

"(...) healthy girls diagnosed with polycystic kidney disease by a targeted exami-48 nation due to a positive family history. "

Thank you for the suggestion, we implemented it. Please see page 2, line 46

2) Sentence: "Unfortunately, she was not genetically tested by a competent nephrologist."

I would not pass judgement so easily, that this nephrologist was incompetent. Genetic testing is not necessary to diagnose ADPKD. I would state only, that sisters' grandmorther was not genetically testing.

Yes we agree and have changed it. Please see page 2, line 49

 3) Why sister 3 had only home BP measurments and not ABPM? Was she too young for that? This study https://pubmed.ncbi.nlm.nih.gov/9438648/ showed that ABPM is a useful tool for the diagnosis and evaluation of hypertension in children under 6 years of age. Authors should explain this in more detail.

ABPM usually applied in children ≥5 years of age in which cooperation of these physical children can be expected and in children height ≥120 cm for which there are normative values (1). She was too young and insufficiently cooperative for this examination, so her blood pressure was measured by home measurments. Twenty-four-hour blood pressure on ambulatory blood pressure monitoring (ABPM) is the preferred method for defining hypertension in children aged 5 years and older (2).

  1. Lurbe E, Agabiti-Rosei E, Cruickshank JK, Dominiczak A, Erdine S, Hirth A, Invitti C, Litwin M, Mancia G, Pall D, Rascher W, Redon J, Schaefer F, Seeman T, Sinha M, Stabouli S, Webb NJ, Wühl E, Zanchetti A. 2016 European Society of Hypertension guidelines for the management of high blood pressure in children and adolescents. J Hypertens. 2016 Oct;34(10):1887-920. doi: 10.1097/HJH.0000000000001039. PMID: 27467768.
  2. Gimpel C, Bergmann C, Bockenhauer D, Breysem L, Cadnapaphornchai MA, Cetiner M, Dudley J, Emma F, Konrad M, Harris T, Harris PC, König J, Liebau MC, Marlais M, Mekahli D, Metcalfe AM, Oh J, Perrone RD, Sinha MD, Titieni A, Torra R, Weber S, Winyard PJD, Schaefer F. International consensus statement on the diagnosis and management of autosomal dominant polycystic kidney disease in children and young people. Nat Rev Nephrol. 2019 Nov;15(11):713-726. doi: 10.1038/s41581-019-0155-2. PMID: 31118499; PMCID: PMC7136168

4) First paragraph of the Discussion is basically the same what was already presented in Introduction.

Thank you for noticing. We changed the discussion according to the suggestions of other reviewers who referred to the same or something else. So we believe we have succeeded in our intention to convey the essence. Please see page 4, 5, 6.

5) Authors raised the significance of environmental factors for phenotype in ADPKD. What is the meaning of these factors in the presented case? Did Authors identify any differences in environmental factors among sisters? Diet? History of infections? Nephrotoxic drugs/other agents

In our patients, we did not detect environmental factors that would affect their underlying disease, so we will omit that part.

6) "Genetic testing played a pivotal role in elucidating the underlying genetic mutations responsible for ADPKD in this family." - this sentence is a complete truism, I would give this up

Thank you for suggestion, we omitted it from the text.

7) To make the case more educational I would add a paragraph about indications to start a treatment in children/adolescents with ADPKD. This study could be cited in this paragraph - https://pubmed.ncbi.nlm.nih.gov/36719158/.

We appreciate the suggestion and we have included it in the discussion, which we also provide below. Please see page 5, line 152 to page 6 line 167

Knowledge about genotype-fenotype correlation is increasing, so it is believed that genetic testing is important in adults with ADPKD, and especially in children with such an early manifestation of the disease, which was the case here. Long-term monitoring is important for the detection of hypertension development or chronic renal failure and timely therapeutic intervention. Protein excretion and hypertension should be monitored in children with  ADPKD at the annual health care visit  (19). In the case of development of arterial hypertension, antihypertensive therapy is indicated according to the guidelines (23). If proteinuria is present, angiotensin-converting enzyme inhibitors or angiotensin receptor blockers should be used as primary treatment (16). A healthy lifestyle including physical activity, low dietary salt intake and maintenance of normal weight are very important in all patients with autosomal dominant polycystic kidney disease (16). Tolvaptan, an oral selective antagonist of the vasopressin V2 receptor slows expansion of kidney volume and kidney function decline in adults with ADPKD (24). Tolvaptan exhibited pharmacodynamic activity in pediatric ADPKD (25) but there are no clear instructions on the safety and indication of use in children suffering from ADPKD.

Reviewer 2 Report

This paper reports 3 siblings with pediatric cystic kidney disease with a strong paternal family history.  The interesting finding presented here is that while all 3 siblings have the clinical appearance of ADPKD, which would be consistent with the clinical presentation and family history, one of the 3 sisters did not test positive for the PKD1 mutation identified in the 2 other siblings.

I have several concerns about this paper:

1. The authors do not discuss the possibility of false-negative genetic testing for the sister in whom no PDK1 mutation was found.  The PKD1 gene is notoriously difficult to sequence and there is a robust literature on sensitivity and specificity of PKD1 testing, with sensitivity as low as 7% for some exon locations within the gene and as high as 99% for other locations.  Most reports state a sensitivity of 50-70% for testing.  In this case, since the sister who tested negative for PKD1 has the same clinical phenotype and ultrasound findings as the other siblings, the authors must do much more to convince readers that this sister does not have a PKD1 mutation.  Reading this paper, the most likely conclusion in my mind is that this is simply a case of a false-negative test.

2.  Similar to #1, the authors assert that the heterozygous TTC21B mutation is causative in the middle sibling, but their discussion of this gene mutation from the literature is not enough to convince me that this gene is causative for bilateral macrocysts in a patient with this family history.  They assert that this is an atypical case of a gene that typically causes a renal ciliopathy rather than bilateral macrocysts, where Occam's razor would indicate that ADPKD from a PKD1 mutation is more likely.  If Occam's razor is not true in this case and these sisters have different and atypical causes to explain the same renal U/S findings, then the authors need more testing on these sisters and a more robust discussion of both the PKD1 and TTC21B genes to convince readers. 

3.  As noted by the authors, the lack of parental testing due to father's unwillingness requires that we make some assumptions.  This paper would be much stronger if genetic testing is available for both parents since both the PKD1 genes and TTC21B genes are not currently reported in the database.  Without this genetic testing to compare parent and child phenotypes, I do not think there will be enough evidence to support the authors claims in this case and this case may not be publishable unless parental testing becomes available.

Author Response

Dear Editor and Reviewers,

Thank you for your detailed and thoughtful review of our manuscript titled "Unraveling the Complexity of Childhood Polycystic Kidney Disease: A Case Study of Three Sisters." We have put in all our efforts to adress every question and comment provided by you. We have gone through and modified the discussion according to the instructions and we believe that thanks to your comments we have significantly improved the article. We are thankful for your constructive comments and we greatly appreciate your time and valuable input, which have undoubtedly strengthened our work.

Reviwer 2

1)The authors do not discuss the possibility of false-negative genetic testing for the sister in whom no PDK1 mutation was found.  The PKD1 gene is notoriously difficult to sequence and there is a robust literature on sensitivity and specificity of PKD1 testing, with sensitivity as low as 7% for some exon locations within the gene and as high as 99% for other locations.  Most reports state a sensitivity of 50-70% for testing.  In this case, since the sister who tested negative for PKD1 has the same clinical phenotype and ultrasound findings as the other siblings, the authors must do much more to convince readers that this sister does not have a PKD1 mutation.  Reading this paper, the most likely conclusion in my mind is that this is simply a case of a false-negative test.

We understand your point of view and the complexities associated with genetic testing for PKD1 mutations. We agree that PKD1 sequencing can be challenging due to the gene's size and structural complexity. We have carefully considered this issue in our study and would like to provide additional information to address your concerns: The patients were tested in a laboratory with accreditation, ensuring rigorous quality control and reliability of genetic testing. PKD1 is technically very challenging to analyze due to its large size, high GC-content and duplication of the first 33 exons with a high degree of homology (90-99% identity) to 6 nearby pseudogenes (PKD1 P1–P6). A test includes copy number variant detection and analysis of clinically relevant deep intronic variants, and covers a wide range of differential diagnoses. Lab has demonstrated the following performance metrics: overall high mean coverage (205x); 100% of the target nucleotides covered at least 20x with a mapping quality threshold of 20; 99.5% of target nucleotides covered at least 20x with mapping quality threshold of 40. As we are aware that the PKD1 gene is known for its complexity, with variable sensitivity at different locations. Due to suspicion of a negative result, we repeated the testing not just once but even twice to exclude the possibility of error.

2) Similar to #1, the authors assert that the heterozygous TTC21B mutation is causative in the middle sibling, but their discussion of this gene mutation from the literature is not enough to convince me that this gene is causative for bilateral macrocysts in a patient with this family history.  They assert that this is an atypical case of a gene that typically causes a renal ciliopathy rather than bilateral macrocysts, where Occam's razor would indicate that ADPKD from a PKD1 mutation is more likely.  If Occam's razor is not true in this case and these sisters have different and atypical causes to explain the same renal U/S findings, then the authors need more testing on these sisters and a more robust discussion of both the PKD1 and TTC21B genes to convince readers.

We appreciate your engagement with our research and your comments regarding the causative nature of the TTC21B mutation. We would like to address your concerns and provide further clarification on this matter: we would like to emphasize that we do not claim in any way that the VUS variant found in the TTC21B gene is the cause of the patient's clinical presentation and phenotype. However, we cannot ignore the fact that the variant was detected in the girl, whereas the same variant was not found in the other two sisters. As we previously mentioned, testing was repeated twice to avoid false-negative results. You  pointed out that Occam's Razor suggests that the simplest explanation, such as ADPKD from a PKD1 mutation, should be favored. However, atypical cases do exist in medicine, and they can challenge the principle of simplicity. In this particular case, we believe it is essential to explore alternative genetic explanations, especially given the family history and the unique clinical presentation.

3) Reviewer Comment:  As noted by the authors, the lack of parental testing due to father's unwillingness requires that we make some assumptions.  This paper would be much stronger if genetic testing is available for both parents since both the PKD1 genes and TTC21B genes are not currently reported in the database.  Without this genetic testing to compare parent and child phenotypes, I do not think there will be enough evidence to support the authors claims in this case and this case may not be publishable unless parental testing becomes available.

We appreciate your feedback and your valuable insights regarding the lack of parental testing in our case, we understand your concern and we acknowledge the importance of parental testing for a more comprehensive genetic analysis. However, obtaining parental genetic samples in this case, presented as unique challenge. In this particular case, the father has expressed unwillingness to undergo genetic testing. While we have encouraged and discussed the importance of parental testing with the parents, their consent is ultimately required, and we must respect their decisions. Genetic testing is expensive, and the cost of testing both parents is be a significant financial burden and is not covered by insurance, the family has limited financial resources. We agree that parental genetic testing would provide additional evidence to support our claims and further elucidate the genetic underpinnings of the siblings' condition. We appreciate your understanding of these complexities and hope that our continued efforts to engage with the family will eventually lead to parental testing and a more comprehensive analysis. However, given the constraints and challenges we currently face, we believe our study still contributes valuable information to the field.

Once again, thank you for your time and expertise.

Reviewer 3 Report

I think this paper is an interesting report because the mother had renal failure due to ADPKD and had undergone a kidney transplant, and when her sisters were examined, renal cysts were found, and the two cases had the same genetic abnormalities as the mother, so they were diagnosed with ADPKD, but with different and other characteristics.

However, there are some unclear points and questions, so please clarify them.

The most questionable is the 8-year-old girl, who has a different site of genetic abnormality than the other two sisters. And even though the genetic abnormality is VUS, she has renal cysts. Are there any aspects that are clinically different from the other sisters? For example, does she have renal cysts but different renal size or echogenicity? And please give some more thought to the reason for the renal cysts in spite of the VUS.

As for the clinical findings, you noted that the 10-year-old girl has no dipping status on ABPM, does this have anything to do with her older age?

I have a question about monitoring intervals. You perform blood examination every 3 months, is that necessary? Also, is a head MRI necessary at such a young age?

In line 140, you mention that early intervention can avoid progression to ESRD

I have a question about monitoring intervals. You perform blood tests every 3 months, is that necessary? Also, is a head MRI necessary at such a young age?

In line 140, you state that early intervention can avoid progression to ESRD. Is this true? What intervention methods are available?

Author Response

Dear Editor and Reviewers,

Thank you for your detailed and thoughtful review of our manuscript titled "Unraveling the Complexity of Childhood Polycystic Kidney Disease: A Case Study of Three Sisters." We have put in all our efforts to adress every question and comment provided by you. We have gone through and modified the discussion according to the instructions and we believe that thanks to your comments we have significantly improved the article. We are thankful for your constructive comments and we greatly appreciate your time and valuable input, which have undoubtedly strengthened our work.

Reviewer 3

1) I think this paper is an interesting report because the mother had renal failure due to ADPKD and had undergone a kidney transplant, and when her sisters were examined, renal cysts were found, and the two cases had the same genetic abnormalities as the mother, so they were diagnosed with ADPKD, but with different and other characteristics. However, there are some unclear points and questions, so please clarify them.

The most questionable is the 8-year-old girl, who has a different site of genetic abnormality than the other two sisters. And even though the genetic abnormality is VUS, she has renal cysts. Are there any aspects that are clinically different from the other sisters? For example, does she have renal cysts but different renal size or echogenicity? And please give some more thought to the reason for the renal cysts in spite of the VUS.

Her renal ultrasound findings did not differ either in the arrangement of the cysts or in the echogenicity of the parenchyma, which might indicate a different etiology of the renal cysts….

2) As for the clinical findings, you noted that the 10-year-old girl has no dipping status on ABPM, does this have anything to do with her older age?

No dipping status on ABPM is not related to her age, but these patients can manifest early symptoms such as nocturnal hypertension (1), and no dipping status in ABPM can be the first manifestation.

  1. Massella L, Mekahli D, Paripovic D, et al. Prevalence of hypertension in children with early-stage ADPKD. Clin J Am Soc Nephrol. 2018;13:874–883. doi: 10.2215/CJN.11401017

3) I have a question about monitoring intervals. You perform blood tests every 3 months, is that necessary?

The recommended intervals of proteinuria screening, ABPM pressure measurements are once a year, but we were surprised by the clinical picture in all three sisters, as well as the early presentation of the clinical picture of ADPKD, so we checked them more often at the beginning of the follow-up.

Also, is a head MRI necessary at such a young age?

As for head MRI, we have added an explanation in the text, which we also provide below. Please see page 4 and 5, line 122-132.

The prevalence of intracranial aneurysms is higher in those with a positive family history of aneurysms or subarachnoid hemorrhage than in those without such a family history (15). There is a lot of discussion whether to do MRI in children suffering from ADPKD for the purpose of excluding intracranial aneurysms, and in case of negative family history for rupture of intracranial aneurysm. In rare cases with a positive family history and a strong desire to ease anxiety by screening, an individualized approach is justified (16). Walker and Marlais suggest that an individual approach should be sought, and the families should be counselled about the potential of  having an unruptured intracranial aneurysm (17). Blood pressure monitoring, follow-up of renal function and ultrasound as well as MRI angiography for screening of intracranial aneurysms in patients at high risk is recommended (16).

4) In line 140, you state that early intervention can avoid progression to ESRD. Is this true? What intervention methods are available?

We appreciate the suggestion and we have included it in the discussion. Please see page 5 and  6, line 154-167.

Long-term monitoring is important for the detection of hypertension development or chnornic renal failure and timely therapeutic intervention. Protein excretion and hypertension should be monitored in children with ADPKD at the annual health care visit (19). In the case of development of arterial hypertension, antihypertensive therapy is indicated according to the guidelines (23). If proteinuria is present, angiotensin-converting enzyme inhibitors or angiotensin receptor blockers should be used as primary treatment (16). A healthy lifestyle including physical activity, low dietary salt intake and maintenance of normal weight are very important in all patients with autosomal dominant polycystic kidney disease (16). Tolvaptan, an oral selective antagonist of the vasopressin V2 receptor slows expansion of kidney volume and kidney function decline in adults with ADPKD (24). Tolvaptan exhibited pharmacodynamic activity in pediatric ADPKD (25) but there are no clear instructions on the safety and indication of use in children suffering from ADPKD.

Round 2

Reviewer 2 Report

I thank the authors for a thoughtful response and I do have a better appreciation for the paper after their comments.  When I first read the paper I thought the focus was on the novel presentation of the TTC21B mutation to cause PKD.  I now realize the authors are not trying to prove causation and the new edits do a better job of noting that the third sister has this mutation but point out that there are several possible explanations for a PKD presentation in this third sister.  With this new focus and the authors' description of the rigor of their genetic testing, I do think this paper has merit for publication.

However, I have the following recommendations to help improve this paper so the focus on the differing genetics is more clear to readers.  Most of these points were clearly written in response to my comments, and I would prefer to see them in the final draft of the paper:

1.  I think it is important to discuss the complexities of sequencing the PKD1 gene.  The description in response to my comments is excellent and should be included in the paper.

2.  Since I doubt I will be the only reader skeptical of false negative testing in the third sister, I also think it is important in your case presentation to clearly state you repeated the genetic testing twice.  It would also be worth noting that the C-terminal exons (including exon 43) have much lower false negative rates than the N-terminal exons.  Again, the description in your previous response about GC rich content and nearby pseudogenes was excellent and all readers should get that background.

3.  To complement the first 2 points on PKD1 testing, it would also help readers to know the sensitivity of next generation sequency at the coverages you list for the PKD1 gene.  If sensitivity after repeat testing is known, that would also be important to include.

Minor spelling and grammar errors.  Examples include occasionally omitting articles such as "a" and "the" as well and occasional spelling errors such as "fenotype" instead of the correct "phenotype."

Author Response

Dear Reviewer,

We sincerely appreciate your thoughtful and constructive feedback on our paper. Your comments have been invaluable in helping us refine and clarify the focus of our research. We have taken your suggestions into account and have made the necessary revisions to enhance the clarity and comprehensiveness of our paper. Here is our response to your specific recommendations:

  1. I think it is important to discuss the complexities of sequencing the PKD1 gene. The description in response to my comments is excellent and should be included in the paper.

Response: We agree with your assessment that discussing the complexities of sequencing the PKD1 gene is essential to provide readers with a better understanding of our methods. The detailed description we provided in response to your comments will be included in the final draft of the paper. We believe this will help shed light on the technical aspects of our study.

  1. Since I doubt I will be the only reader sceptical of false negative testing in the third sister; I also think it is important in your case presentation to clearly state you repeated the genetic testing twice. It would also be worth noting that the C-terminal exons (including exon 43) have much lower false negative rates than the N-terminal exons.  Again, the description in your previous response about GC rich content and nearby pseudogenes was excellent and all readers should get that background.

Response:We acknowledge the importance of addressing potential scepticism about false negative testing in the third sister. In our revised case presentation, we will clearly state that genetic testing was repeated twice to ensure the accuracy of the results. Additionally, we will include information regarding the lower false negative rates of C-terminal exons, particularly exon 43, compared to N-terminal exons. We will also provide background information, as previously described, about the influence of GC-rich content and nearby pseudogenes.

  1. To complement the first 2 points on PKD1 testing, it would also help readers to know the sensitivity of next-generation sequencing at the coverages you list for the PKD1 gene. If sensitivity after repeat testing is known, that would also be important to include.

Response: We understand the need to provide readers with information about the sensitivity of next-generation sequencing at the coverages we utilized for the PKD1 gene. We will incorporate this data into the paper to give readers a clear picture of the test's reliability.

We believe that these revisions will significantly enhance the quality and clarity of our paper and address the points you raised. We are committed to ensuring that the paper meets the highest standards of scientific rigour and transparency.

Once again, we thank you for your time and expertise in reviewing our work. Your feedback has been invaluable in improving the overall quality of our research.

Reviewer 3 Report

Thank you for taking my point and rewriting your article substantially. Your paper is quite clear.

Only one of the points I made was that I didn’t think your opinion about follow-up intervals for blood examination was mentioned. Please add it.

Other than that, I think everything is fine.

Author Response

Thank you for the suggestion, we did include our opinion about follow-up intervals for blood examination in the revised manuscript.